# The C4 Protein from Tomato Yellow Leaf Curl Virus Can Broadly Interact with Plant Receptor-Like Kinases

**DOI:** 10.3390/v11111009

**Published:** 2019-10-31

**Authors:** Borja Garnelo Gómez, Dan Zhang, Tábata Rosas-Díaz, Yali Wei, Alberto P. Macho, Rosa Lozano-Durán

**Affiliations:** 1Shanghai Center for Plant Stress Biology, Center for Excellence in Molecular Plant Sciences, Chinese Academy of Sciences, Shanghai 201602, China; borja@psc.ac.cn (B.G.G.); zhangdan@sibs.ac.cn (D.Z.); Tabatarosas@uma.es (T.R.-D.); weiyali@sibs.ac.cn (Y.W.);; 2University of the Chinese Academy of Sciences, Beijing 100049, China

**Keywords:** geminivirus, C4, TYLCV, RLK, CLV1, FLS2, BRI1, BRL3, NIK1, PSKR1

## Abstract

Plant receptor-like kinases (RLKs) exert an essential function in the transduction of signals from the cell exterior to the cell interior, acting as important regulators of plant development and responses to environmental conditions. A growing body of evidence suggests that RLKs may play relevant roles in plant-virus interactions, although the details and diversity of effects and underlying mechanisms remain elusive. The C4 protein from different geminiviruses has been found to interact with RLKs in the CLAVATA 1 (CLV1) clade. However, whether C4 can interact with RLKs in other subfamilies and, if so, what the biological impact of such interactions might be, is currently unknown. In this work, we explore the interaction landscape of C4 from the geminivirus Tomato yellow leaf curl virus within the Arabidopsis RLK family. Our results show that C4 can interact with RLKs from different subfamilies including, but not restricted to, members of the CLV1 clade. Functional analyses of the interaction of C4 with two well-characterized RLKs, FLAGELLIN SENSING 2 (FLS2) and BRASSINOSTEROID INSENSITIVE 1 (BRI1), indicate that C4 might affect some, but not all, RLK-derived outputs. The results presented here offer novel insight on the interface between RLK signaling and the infection by geminiviruses, and point at C4 as a potential broad manipulator of RLK-mediated signaling.

## 1. Introduction

Receptor kinases (or receptor-like kinases (RLKs) in plants) are transmembrane proteins localized at the surface of eukaryotic cells, containing an extracellular domain (ECD), a transmembrane domain (TMD), and an intracellular kinase domain (KD). In plants, RLKs play a crucial role in the transduction of signals from the cell exterior to the cell interior, regulating a plethora of different processes during development, as well as during the interaction of plants with their environment [1,2,3]. This functional diversity is enabled by the large expansion of the RLK family, which comprises more than 600 members in Arabidopsis thaliana (hereafter referred to as Arabidopsis) and more than 1000 members in rice [4].

The ECDs of RLKs are diverse and can contain different domains, including leucine-rich repeats (LRR), extensin-like, lectin-like, epidermal-growth-factor-like repeats, and LysM, among others [4]. ECDs bind extracellular ligands of endogenous or exogenous origin, such as peptides, steroids, and saccharides, and can mediate homo- or hetero-dimerization of RLKs [1,2,3,5]. Upon perception of the corresponding ligand, the intracellular KD associates with interacting partners to initiate signal relay inside the cell.

Given the multifaceted nature of plant-pathogen interactions, it is not surprising that RLKs can influence this interface at multiple levels. First, and most evidently, some RLKs act as pattern-recognition receptors (PRRs), mediating perception of molecular patterns from pathogens (pathogen-associated molecular patterns (PAMPs)) or produced by the plant upon recognition of a biotic threat (damage-associated molecular patterns (DAMPs)) [6]. PRRs then initiate a signaling cascade that leads to the activation of pattern-triggered immunity (PTI). Non-PRR RLKs may also regulate other aspects of plant defense, such as the intercellular movement of RNA silencing [7]. In addition, RLKs mediating growth and developmental processes, like the receptor of the plant steroid hormone brassinosteroid (BR), BRASSINOSTEROID INSENSITIVE 1 (BRI1), could indirectly affect defense outputs through molecular cross-talks [8] or influence pathogen performance in other defense-independent ways.Although the relevance of RLKs, and in particular PRRs, in plant interactions with extracellular pathogens such as bacteria and fungi is uncontested, their involvement in plant-virus interactions is currently controversial (reviewed in [9,10]). PRR RLKs have been shown to influence the outcome of viral infections at least in certain RNA virus-host combinations [11,12], although the nature of the putative ligand(s) remains enigmatic. Other RLKs, not presently described as PRRs, have also been found to play a role in anti-viral defense. This is the case of NSP-INTERACTING KINASE (NIK1), which inhibits translation of viral genes in infections by the DNA viruses geminiviruses, and of BARELY ANY MERISTEM 1 (BAM1), which promotes the intercellular spread of RNA silencing [7,13]. Interestingly, enhancing BR signaling, which depends on BRI1, dramatically alleviates symptom development in tomato plants infected with a geminivirus [14].

In recent years, several cases of geminivirus-encoded proteins targeting RLKs have been documented, underscoring the relevance of this family of receptors for the geminiviral infection. The nuclear shuttle protein (NSP) of the bipartite geminiviruses Tomato golden mosaic virus (TGMV), Tomato crinkle leaf yellow virus (TCrLYV), and Cabbage leaf curl virus (CaLCuV) interacts with the intracellular domain of Arabidopsis NIK1 (and its homologues NIK2 and NIK3) [15], inhibiting its function in anti-viral defense [13,16,17]. C4/AC4 from different geminiviruses has been found to interact with RLKs in the CLAVATA 1 (CLV1) clade (reviewed in [10,18]): C4 from Mungbean yellow mosaic virus (MYMV) interacts with Arabidopsis BARELY ANY MERISTEM 1 (AtBAM1) [19]; C4 from Tomato yellow leaf curl virus (TYLCV) interacts with AtBAM1, its close homologue AtBAM2, and with BAM1 orthologues in tomato and Nicotiana benthamiana [7]; and C4 from Beet severe curly top virus (BSCTV) interacts with tomato CLV1 [20]. However, whether C4 can interact with RLKs outside of the CLV1 clade and, if so, what the biological impact and relevance of such interactions would be, remain open questions.

In this work, we explore the interaction landscape of C4 from TYLCV within the Arabidopsis RLK family. For this purpose, we selected eight RLKs from Arabidopsis, spanning five different subfamilies, and tested the interaction with C4 in a targeted manner through two independent methods in planta, namely co-immunoprecipitation (co-IP) and Fӧrster resonance energy transfer-fluorescence lifetime imaging microscopy (FRET-FLIM). Our results indicate that C4 can broadly interact with plant RLKs and that these interactions cannot be predicted by the subfamily classification of RLK family members. Functional analyses of the interaction of C4 with two well-characterized RLKs, the PRR FLAGELLIN SENSING 2 (FLS2) and the non-PRR BRI1, revealed that C4 might affect some, but not all, RLK-derived outputs. Our results shed new light on the interface between RLK signaling and the infection by geminiviruses and suggest C4 as a potential broad manipulator of plasma membrane-transduced signaling.

## 2. Materials and Methods

### 2.1. Plant Material

All transgenic Arabidopsis plants used in this work are in the Col-0 background. 35S:C4 (L5 and L7) and 35S:GFP transgenic lines are described in [7]. T2 35S:C4 L5 plants were used in Figure 4A,B,E; T3 35S:C4 L5 and L7 plants were used in Figure 4C,D,F.

### 2.2. Plasmids and Cloning

Constructs for co-IP and FRET-FLIM assays were generated using Gateway technology (Life Technologies, Carlsbad, CA, USA). C4-GFP, BAM1-RFP, and NIK1-RFP constructs are already described [7]. The CLV1, BRI1, BRL3, PSKR1, FLS2, BAK1, and PERK1 open reading frames (ORFs) were amplified from Arabidopsis cDNA with primers listed in Appendix A. CLV1, BRL3, FLS2, and BAK1 entry clones were generated by cloning into pENTR^™^ TOPO^®^ (Invitrogen, Carlsbad, CA, USA). BRI1, PSKR1, and PERK1 entry clones were generated by cloning into pENTR/pDONR207 (Invitrogen) vector by Gateway^®^ BP clonase reaction (Invitrogen). All ORFs were subsequently subcloned into the expression vector pB7WRG2.0 (-RFP) [21] by Gateway^®^ LR Recombination reaction (Invitrogen) following the manufacturer’s instructions.

### 2.3. Transient Expression in N. benthamiana

Transient co-expression was performed in three-week-old to four-week-old N. benthamiana leaves through Agrobacterium infiltration (OD600 = 0.5). For both co-IP and FRET-FLIM assays, a clone to express C4-GFP [7] was co-infiltrated with clones to express BAM1-, CLV1-, BRI1-, BRL3-, PSKR1-, FLS2-, NIK1-, BAK1-, and PERK1-RFP (see “Plasmids and cloning”). Samples were taken two days after infiltration.

### 2.4. Protein Extraction and Co-Immunoprecipitation

Two days after infiltration, 0.75–1 g of infiltrated N. benthamiana leaves were harvested. Protein extraction, co-immunoprecipitation (co-IP), and western blot were performed as previously described [22]. For western blot, the following primary and secondary antibodies were used: mouse anti-green fluorescent protein (GFP) (M0802-3a, Abiocode, Agoura Hills, CA, USA) (1:10000), rat anti-red fluorescent protein (RFP) (5F8, Chromotek, Planegg-Martinsried, Germany) (1:10000), goat polyclonal anti-mouse coupled to horseradish peroxidase (Sigma, St. Louis, MO, USA) (1:15000), and goat polyclonal anti-rat coupled to horseradish peroxidase (Abcam, Cambridge, UK) (1:15000).

### 2.5. FRET-FLIM

N. benthamiana leaf discs transiently co-expressing the proteins of interest were used to perform FRET-FLIM experiments. C4–GFP was used as a donor protein whereas the different RLKs fused to RFP were used as acceptor proteins. FRET-FLIM analysis was performed as described previously [7].

### 2.6. Hormone and Peptide Treatments

For gene expression analysis after flg22 and BL treatment, transgenic 35S:GFP or 35S:C4 Arabidopsis seedlings [7] were sown in half-strength Murashige and Skoog (MS) medium. After six days, three technical replicates of four independent experimental sets (mock-flg22, flg22, mock-PPZ-BL, and PPZ-BL) with four to six seedlings each were transferred to liquid half-strength MS medium. After three days, the medium of PPZ-BL and mock-PPZ-BL sets was supplied with 2 µM Propiconazole (PPZ) or 2 µM DMSO. Six days after transfer and one hour prior to harvesting, PPZ-BL and mock-PPZ-BL were supplied with 1 µM epiBrassinolide (epiBL) and 1 µM 80% ethanol (v/v), respectively. At the same time point, flg22 and mock-flg22 sets were treated with 1 µM flg22 and 1 µM distilled water, respectively.

For seedling growth inhibition upon flg22 treatment, seedlings were sown in half-strength MS medium supplied with 0.9% phytoagar. After four days, at least 12 seedlings per line were transferred individually to each well of a 48-well plate containing 500 µL of half-strength MS supplied with 1% sucrose and 100 nM flg22. Seedlings were grown in light for 10 days and weighed after blotted dry.

For BL root growth response assays, seedlings were sown in half-strength MS medium supplied with 0.9% phytoagar. After six days, seedlings were transferred to new MS medium supplied with 2 µM propiconazole (PPZ) with or without epiBrassinolide (0.1 or 1 nM). Root length was measured six days later.

### 2.7. RNA Extraction and qRT-PCR

To study gene expression after flg22 and BL treatment (see previous section for further details), four to six frozen 12-day-old transgenic 35S:GFP or 35S:C4 Arabidopsis seedlings [7] were ground and RNA purification was performed with the OMEGA BIOTEK kit (OMEGA BIOTEK, Norcross, GA, USA) following the manufacturer’s instructions. DNAase treatment and cDNA synthesis were done with the iSript gDNA clear cDNA synthesis kit (BIO-RAD, Hercules, CA, USA). Quantitative reverse transcription-polymerase chain reaction (RT-PCR) was done as previously described [23] using primers listed in Appendix A. ACTIN (ACT2) was used as normalizer.

### 2.8. Detection of Reactive Oxygen Species (ROS)

Four-week-old to five-week-old wild-type and 35S:C4 Arabidopsis plants grown in short day conditions (10 h light/14 h dark cycle) were used to detect the ROS burst upon flg22 treatment as previously described [24]. Three leaf disks of at least eight plants per genotype were individually collected on a white 96-well plate (OptiPlateTM-96, PerkinElmer, Waltham, MA, USA) filled with 100 µl of distilled water and incubated for 14–16 h in light. Water was carefully removed from each well and replaced by 100 µL of elicitor mix containing 100 nM flg22, 100 µM luminol, and 20 μg/mL HRP. ROS production was monitored over 60 min using a microplate reader (Thermo Varioskan Flash, ThermoFisher, Waltham, MA, USA).

### 2.9. Phylogeny and Protein Sequence Alignment

Protein sequences were retrieved from the NCBI database. Phylogenetic analysis and protein sequence alignment were performed using CLC Workbench 10. A Clustal W multiple alignment of amino acid sequences was performed with a BLOSUM cost matrix with a gap open cost of 10 and a gap extend cost of 0.1. Protein alignment was used as base for building the tree in which genetic distances were analyzed under the Jukes-Canto model with a Neighbor-Joining method without using an outgroup.

## 3. Results and Discussion

A growing body of evidence has made it clear that the geminivirus-encoded C4 protein can interact in planta with RLKs from the CLV1 clade, which has been associated with the promotion of virulence and symptom development [7,19,20]. However, whether C4 can associate with RLKs from other subfamilies with more divergent intracellular domains and if this could have an impact on the outcome of the viral infection is currently unknown.

With the aim to explore the ability of C4 to interact with other members of the RLK family, we selected eight RLKs spanning five families: BAK1 and NIK1 from LRR II; BRI1, BRI1-LIKE 3 (BRL3), and PHYTOSULFOKIN RECEPTOR 1 (PSKR1) from LRR X; CLV1 from LRR XI; FLS2 from LRR XII; and PROLINE-RICH EXTENSIN-LIKE RECEPTOR KINASE 1 (PERK1) from PERKL (Figure 1). This list comprises both RLKs described as PRRs (BAK1, FLS2) and not described as PRRs (others). These selected RLKs were cloned, and their accumulation and plasma membrane localization upon transient expression in N. benthamiana were confirmed.

Co-IP identifies proteins associated with the immunopurified protein of interest, whether this interaction is direct or indirect. The selected RLKs fused to RFP at their C-terminus were transiently co-expressed with C4 fused to GFP in N. benthamiana leaves and subjected to co-IP assays. As shown in Figure 2A,B, CLV1-, BRI1-, BRL3-, PSRK1-, FLS2-, and NIK1-RFP were detected as associated with C4, while BAK1- and PERK1-RFP were not.

As opposed to co-IP, the measurable energy transfer in FRET-FLIM requires close proximity [25], therefore detecting interactions that are mostly direct. Consistently with the co-IP results, BAK1-RFP and PERK1-RFP did not show an interaction with C4-GFP in FRET-FLIM assays, as reflected by the lack of reduction in the GFP lifetime. Co-expression of CLV1-, BRI1-, BRL3-, PSKR1-, FLS2-, or NIK1-RFP, however, produced a statistically significant reduction in the lifetime of C4-GFP (Figure 2C), indicative of a positive interaction. Therefore, C4 can interact with CLV1, BRI1, BRL3, PSKR1, FLS2, and NIK1 in vivo, as indicated by co-IP and FRET-FLIM, and this interaction is likely to be direct.

Interestingly, the behavior of the selected RLKs in their association with C4 does not follow their phylogenetic distribution (Figure 1 and Figure 2). With the aim of identifying features that may determine the ability of a given RLK to interact with this viral protein, we aligned the cytoplasmic domains of the selected RLKs, separating those which interact from those which do not. This comparison, however, did not provide any clear conclusion, since no residue could be identified as differential and conserved between and within interactors and non-interactors (Appendix A), suggesting that the binding may rely more on tertiary rather than primary structure.

Our results indicate that C4 from TYLCV can interact broadly with members of the RLK family (Appendix A). It has to be considered, however, that TYLCV is a phloem-restricted virus. Therefore, during the natural infection, C4 will only have access to those RLKs that are expressed in this tissue. In order to identify potentially C4 targeted-RLKs relevant in the context of the viral infection, we analyzed the expression of the genes encoding the selected RLKs in the phloem according to two published datasets [26,27] (Figure 3). Although there are differences in gene expression between the two datasets analyzed, as shown in Figure 3, all tested RLKs were expressed in the phloem in both cases, and therefore their coexistence with C4 in the cell is likely. Considering this, it is plausible that the positive interactions identified in this work have biological significance, although this possibility needs to be experimentally investigated.

The interaction between C4 and BAM1 has been previously proposed to likely affect some, but not all, BAM1-mediated functions. This is based on the observation that C4-expressing plants display a reduced intercellular movement of RNAi, which requires BAM1 and BAM2, but responds normally to CLV3 peptide in root growth assays, a response that depends on BAM1 [7]. With the aim to investigate whether C4 affects the functionality of other interacting RLKs, we decided to test whether FLS2- and BRI1-dependent responses to the immunogenic bacterial peptide flg22 or to the steroid hormone epibrassinolide (epiBL), for which they are the respective receptors, are altered in transgenic C4-expressing Arabidopsis lines. For this purpose, we selected well-known readouts of each of these pathways: gene expression (FRK1, CYP8IF2), production of an apoplastic burst of ROS, inhibition of seedling growth upon flg22 treatment to evaluate FLS2 function; gene expression (EXP8) and root growth following exogenous epiBL treatments to evaluate BRI1 function. The results of these functional analyses are shown in Figure 4. Although the early apoplastic ROS burst following flg22 perception was reduced by the presence of C4, the intermediate activation of marker gene expression or the late inhibition of growth occurred normally in the C4-expressing plants (Figure 4A–C). Strikingly, the expression of the marker gene EXP8, which is activated upon BL perception, remained at basal levels in epiBL-treated C4-expressing plants, in contrast to the marked induction in control plants (Figure 4D). Despite the dramatic effect on the expression of this marker gene, C4 did not affect root growth responses to epiBL (Figure 4E). Although it has to be noted that indirect effects of C4 on the FLS2- and BRI1-dependent pathways, not associated to its physical interaction with the respective RLKs, cannot be ruled out, our results indicate that C4 may affect some, but not all, readouts downstream of RLK function, directly or indirectly. Since C4 interacts with the kinase domain of BAM1 [7], one possibility would be that the viral protein exerts a negative impact on the kinase activity of the RLKs. However, such an inhibition of the enzymatic activity would be expected to abolish downstream events following perception of the extracellular ligand, which is not the case for all outputs tested following interaction of C4 with BAM1 [7], FLS2, or BRI1. We therefore hypothesize that a more likely mechanism of action would be the displacement of endogenous interactors by C4, although this idea awaits experimental testing.

Two different scenarios can be envisioned to justify the broad capacity of C4 to interact with different, unrelated RLKs. One possibility is that these RLKs play a role in the interaction between plant and virus and are therefore bona fide targets that C4 has evolved to bind and manipulate. The alternative possibility is that only one or some RLKs are bona fide targets, while the interaction with the others is merely a side effect of the evolved capacity of C4 to bind the former. The identification of the structural features that determine the interaction with C4 might shed light on this issue. Although all identified interacting RLKs are expressed in the phloem and hence have the potential to impact the infection by TYLCV, their functional relevance during this process remains unexplored.

Mounting evidence suggests that PRRs might play a role also in plant-virus interactions. Although this has not been proven for geminiviruses yet, TYLCV infection can activate plant defense responses [28,29], and activation of PTI by flg22 treatment can trigger the release of C4 from the plasma membrane and its translocation to the chloroplast, suggesting that this viral protein is somehow associated with and responsive to the activation of the FLS2 receptor complex (Medina-Puche et al., unpublished). To date, however, NIK1 is the only RLK shown to have a prominent role in the geminivirus infection. The finding that C4 can physically interact with NIK1 raises the enticing idea that C4 might be exerting, in monopartite geminiviruses, the function of the NSP of bipartite viruses in suppressing the anti-viral translational inhibition [13], although further experiments will be required to determine whether this is the case.

## Figures and Tables

**Figure 1 viruses-11-01009-f001:**
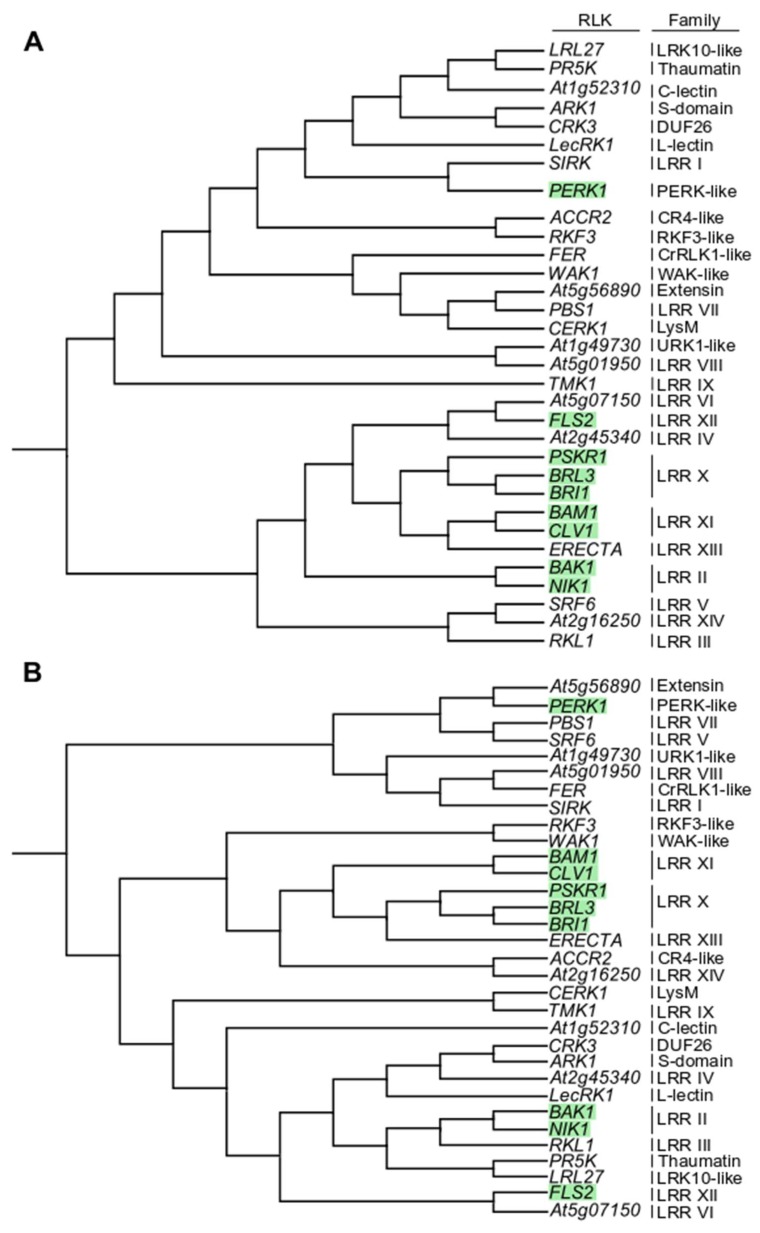
Phylogeny of representative members of the Arabidopsis receptor-like kinase (RLK) family. (**A**) Cladogram of representative Arabidopsis RLKs based on the full protein alignment. (**B**) Cladogram of representative Arabidopsis RLKs based exclusively on the intracellular domain. Green squares highlight RLKs used in this study. Families are indicated on the right. The phylogenetic analysis was performed using CLC Workbench 10.

**Figure 2 viruses-11-01009-f002:**
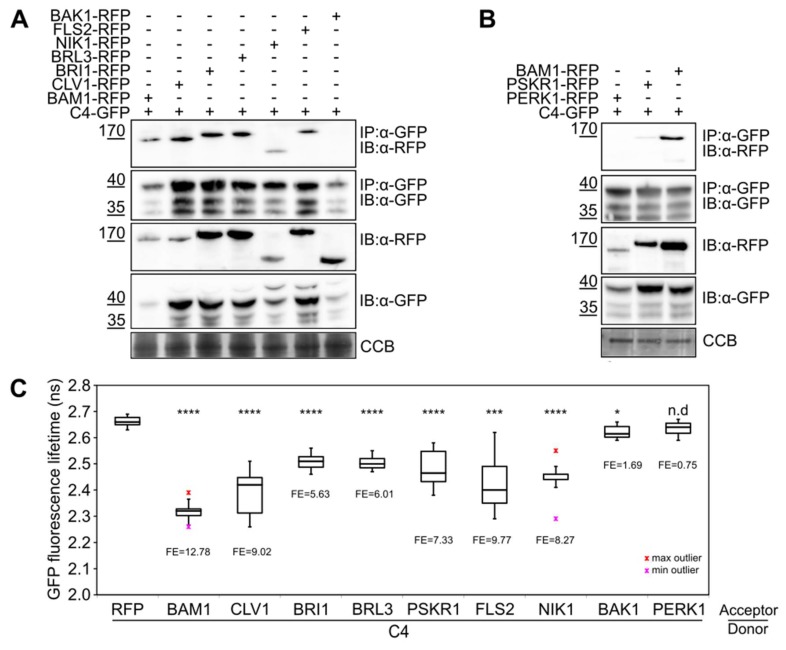
C4 interacts with several leucine-rich repeat (LRR)-RLKs in vivo. (**A**) Co-immunoprecipitation of BAM1-, CLV1-, BRI1-, BRL3-, NIK1-, FLS2-, and BAK1-RFP with C4-GFP following transient expression in N. benthamiana. (**B**) Co-immunoprecipitation of BAM1-, PSKR1- and PERK1-RFP with C4-GFP following transient expression in N. benthamiana. IP: immunopurification, IB: immunoblotting. Three independent biological replicates were performed with similar results; one replicate is shown. (**C**) Fӧrster resonance energy transfer-fluorescence lifetime imaging microscopy (FRET-FLIM) analysis of the RLKs-RFP/C4-GFP interaction following transient expression in N. benthamiana leaves. Box plots denote distribution of eight measurements ± SD. FE: FRET Efficiency. Asterisks indicate a statistically significant difference according to Student’s t-test (**** p value < 0.0001; *** p value < 0,005; * p value < 0.05; n.d, no difference). Colored markers above and below whiskers indicate outliers. Three independent biological replicates were performed with similar results. One replicate is shown.

**Figure 3 viruses-11-01009-f003:**
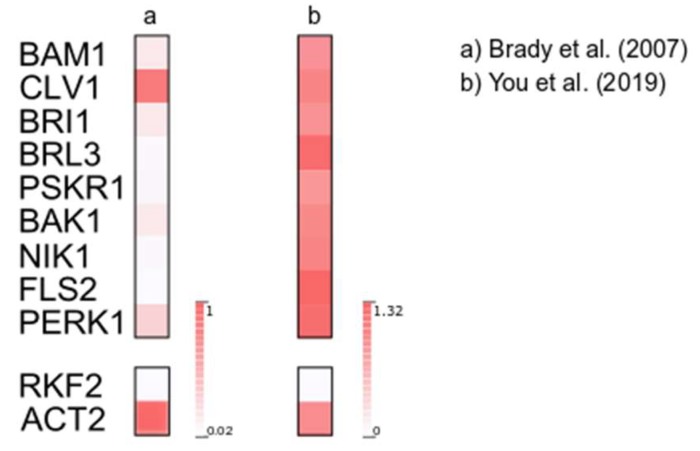
C4-interacting RLKs are expressed in phloem companion cells. Expression of selected RLKs according to publicly available transcriptome datasets [26,27]. For (**a**), mean expression values were selected from the SUC2:GFP-marked cell population dataset, in which companion cells are overrepresented [26]; for (**b**), mean expression values were selected from companion cells read counts in mock conditions [27]. All values are normalized to ACT2 expression in the same experiment. Expression of the pollen-specific RECEPTOR-LIKE SERINE/THREONINE KINASE 2 (RKF2) is shown as negative control.

**Figure 4 viruses-11-01009-f004:**
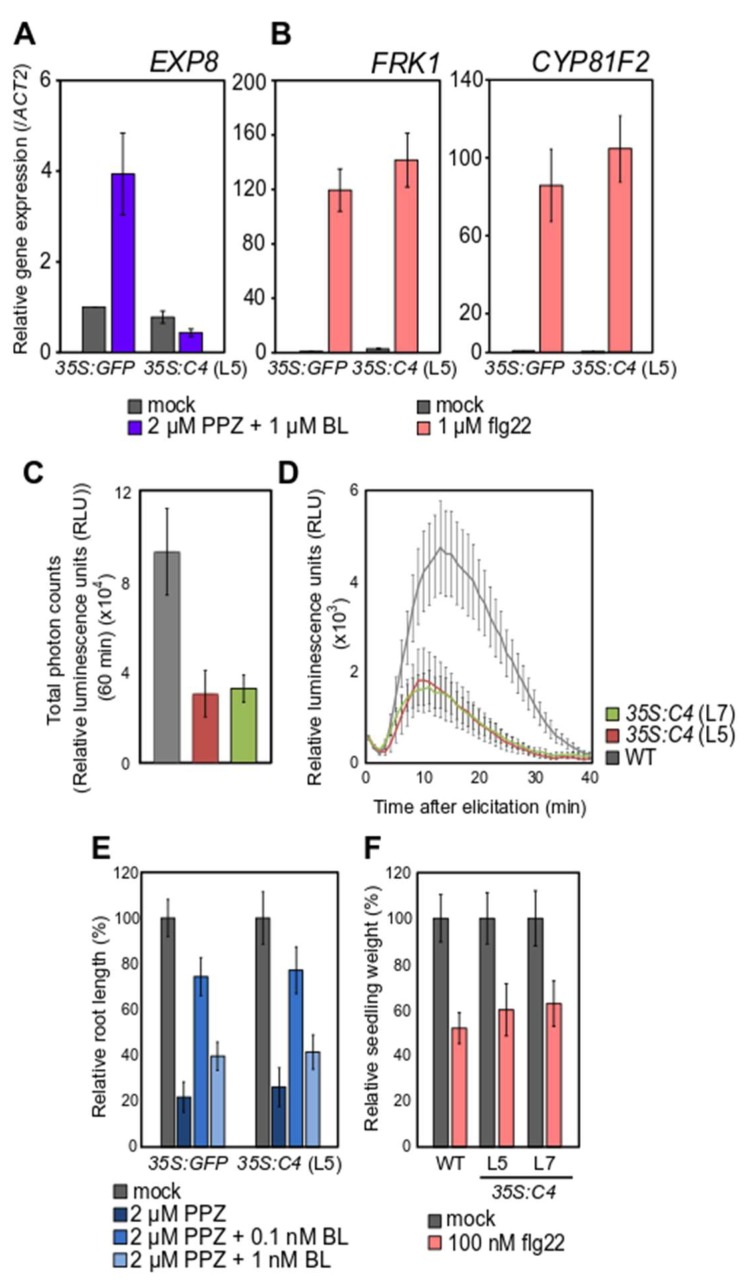
C4 affects some, but not all, BRI1 and FLS2 downstream readouts. Relative expression of the BL responsive gene EXP8 (**A**) or of the flg22 responsive genes FRK1 and CYP8IF2 (**B**). Bars represent mean expression in 10-day-old T3 35S:GFP and T2 35S:C4 (L5) transgenic Arabidopsis seedlings treated with 1 µM epiBL (**A**) or 1 µM flg22 (**B**) one hour prior sample harvesting ± SD, n = 3 (technical replicates), as measured by quantitative real-time-polymerase chain reaction (qRT-PCR). Three independent biological replicates were performed, and results from one representative replicate are shown. (**C**,**D**) Production of reactive oxygen species (ROS) in four-week-old wild-type (WT) and T3 35S:C4 transgenic Arabidopsis (L5, L7) plants upon 100 nM flg22 treatment measured as total photon counts during 60 min (**C**) (bars represent averages values ± SE, n = 12) or as relative luminescence units (RLU) along 40 min (**D**). Three independent biological replicates were performed; values correspond to one representative replicate ± SE, n = 12 (leaf discs from independent leaves). (**E**) Normalized root length of 12-day-old T3 35S:GFP and T2 35S:C4 (L5) transgenic Arabidopsis seedlings following depletion of (2 µM Propiconazol (PPZ)) and exogenous addition of brassinosteroids (0.1 and 1 nM epiBL) for six days, ±SD, n = 12–16. Three independent biological replicates were performed with similar results; results from one replicate are shown. (**F**) Relative seedling growth inhibition of 10-day-old WT and T3 35S:C4 transgenic Arabidopsis (L7, L5) seedlings upon 100 nM flg22 treatment ± SE, n = 14–16. Three independent biological replicates were performed with similar results; results from one replicate are shown.

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
