# Peer review of "The C4 Protein from Tomato Yellow Leaf Curl Virus Can Broadly Interact with Plant Receptor-Like Kinases"

_viruses, 2019, doi:10.3390/v11111009_

Round 1

Reviewer 1 Report

General comments:

The manuscript “The C4 protein from Tomato yellow leaf curl virus can broadly interact with plant receptor-like kinase” by Gómez et al. reports that the C4 protein of Tomato yellow leaf curl virus (TYLCV) can broadly interact with 6 plant receptor-like kinases (CLV1, BRI1, BRL3, PSKR1, FLS2 and NIK1). In this study, the authors tried to identify the RLK subfamily, besides CLV1, that can interact with the C4 protein. They classified Arabidopsis RLKs based on protein sequence alignment. They selected 9 RLKs from 6 different subfamilies and performed co-IP and FRET-FLIM assay using the C4 protein. These 6 RLKs were also expressed in the phloem companion cells, according to database analysis. Gene expression analysis of the epibrassinolide (epiBL) marker-coding genes showed that EXP8 gene was reduced, and FRK1 and CYP8IF2 were not affected in C4 overexpressed transgenic Arabidopsis. The authors concluded that the C4 protein of TYLCV can interact with RLKs in different subfamilies and affect part readouts of brassinosteroid-mediated pathway. The information of the manuscript is worth publishing, however, some additional work and changes are required to improve the manuscript.

Major suggestions:

Throughout the text, the authors tend to use long sentences consisting of several points in a single sentence, which is confusing and difficult to read. For example: page 1, line 36-41; this sentence covers 4 things.

RLKs play a crucial role in signal transduction systems. These signal transduction systems regulate different biological processes including plant development and response to environment. The functional diversity is enabled by the large expansion of the RLK family. The RLK family comprises more than 600 members in Arabidopsis and more than 1000 members in rice.Similar cases can be found in line 49-53, line 207-210, and line 248-252.

This sentence is complex and the major point of the statement is unclear and difficult to understand. Is there a conserved domain among different RLKs that interact with C4 protein? It may provide information for further study to illustrate the relationship between C4 and interacting RLKs. Most conclusions in this study relied on speculation without actual verification. Although the interaction between C4 and CLV1, BRI1, BRL3, PSKR1, FLS2 and NIK1 were proved by co-IP and FRET in this study, it is unclear whether or not C4 involves in signal transduction or gene regulation. The statement “C4 as a potential broad manipulator of RLK-mediated signaling” is not suitable. The only experiment about the roles of C4 participating in different RLKs involved pathway is epiBL treatment analysis. However, the results did not show clear evidence whether C4 involves in epiBL-mediated pathway. These C4-RLKs interactions may occur because of 3rd or 4th structure for physical contact, and these interactions may not indicate C4 is involved in pathway regulation. The effect of EXP8 gene expression may be due to other biological process as well but not due to C4-RLKs (FLS2 or BRI1) interaction.

Minor suggestions:

Page 1, Line 42: typo “ECDs”

Page 1, Line 44: typo “ECDs”

Page3, Line 124: (Materials and methods for hormone treatment) how many biological repeats were included in this assay?

Page 7, Line 253: The authors mentioned “6 subfamilies” in the text, and the classification should be indicated in the result of phylogenetic analysis (Figure 1).

Page 10, Line 278: What is the relation between transgenic plants used figure 4A-B and 4C-D? Is the result of A-B derived from only one transgenic line? It must be described in the materials, results and figure legend.

Author Response

We would like to thank the reviewers for their critical assessment of our manuscript, and provide a point-by-point response to their comments below.

REVIEWER 1 

Major suggestions:

Throughout the text, the authors tend to use long sentences consisting of several points in a single sentence, which is confusing and difficult to read. For example: page 1, line 36-41; this sentence covers 4 things.

RLKs play a crucial role in signal transduction systems. These signal transduction systems regulate different biological processes including plant development and response to environment. The functional diversity is enabled by the large expansion of the RLK family. The RLK family comprises more than 600 members in Arabidopsis and more than 1000 members in rice.Similar cases can be found in line 49-53, line 207-210, and line 248-252.

We thank the reviewer for this suggestion. We have now simplified the indicated sentences by splitting them in two or more (see lines 36-41, 49-53, 209-212, and 251-255).

Is there a conserved domain among different RLKs that interact with C4 protein? It may provide information for further study to illustrate the relationship between C4 and interacting RLKs. Most conclusions in this study relied on speculation without actual verification. Although the interaction between C4 and CLV1, BRI1, BRL3, PSKR1, FLS2 and NIK1 were proved by co-IP and FRET in this study, it is unclear whether or not C4 involves in signal transduction or gene regulation. The statement “C4 as a potential broad manipulator of RLK-mediated signaling” is not suitable. The only experiment about the roles of C4 participating in different RLKs involved pathway is epiBL treatment analysis. However, the results did not show clear evidence whether C4 involves in epiBL-mediated pathway. These C4-RLKs interactions may occur because of 3rd or 4th structure for physical contact, and these interactions may not indicate C4 is involved in pathway regulation. The effect of EXP8 gene expression may be due to other biological process as well but not due to C4-RLKs (FLS2 or BRI1) interaction.

As suggested by the reviewer, we had indeed tried to analyze the predicted structure of the kinase domains of the selected RLKs, modelling them on the resolved BRI1 structure. However, no obvious structural difference correlating with the interaction with C4 emerged. Hopefully, mapping of the interaction coupled to an increase in the availability of structural information will help shed light on this matter in the future. We absolutely agree with the reviewer in that an effect of C4 on RLK-mediated pathways through the interaction with the receptors is currently speculative. In this manuscript, we aimed at showing that C4 can interact with different RLKs, which is something that, in our opinion, should be considered when studying the function of this protein during the viral infection. Whether these physical interactions have an impact in the corresponding downstream signalling cascades will require further investigation. In order to get some hints regarding whether C4 could completely abolish RLK function through physical association to the intracellular domain of these proteins, we took advantage of the well-characterized RLKs FLS2 and BRI1, both of which are bound by C4. Using treatments with the corresponding ligands (flg22 and epiBL, respectively) and known readouts for these signalling pathways, we could demonstrate that the presence of C4 does not affect all responses downstream of RLK activation. We do see an impact of C4 on ROS production (upon flg22 treatment) and EXP8 expression (upon epiBL treatment), although the direct or indirect nature of these effects is at present unknown, as pointed out by the reviewer. Indeed, we do not claim that C4 is a broad manipulator of RLK signalling, but that it has the potentiality to be, since it associates with different RLKs. We have carefully phrased the text in the manuscript to try to reflect these facts more clearly and not mislead the reader (see, for example, lines 27-30; lines 91-92; lines 227-230).

Minor suggestions:

Page 1, Line 42: typo “ECDs”

We thank the reviewer for spotting this mistake, which has now been corrected.

Page 1, Line 44: typo “ECDs”

We thank the reviewer for spotting this mistake, which has now been corrected.

Page3, Line 124: (Materials and methods for hormone treatment) how many biological repeats were included in this assay?

Three independent biological replicates were performed for this assay; this is indicated in the corresponding figure legend (lines 285-286).

Page 7, Line 253: The authors mentioned “6 subfamilies” in the text, and the classification should be indicated in the result of phylogenetic analysis (Figure 1).

We apologize for this mistake: it should read “five subfamilies” – this has now been corrected. The families the different RLKs in the phylogenetic tree belong to are indicated on the right – this has now been specified in the figure legend, and the corresponding label has been added to the figure.

Page 10, Line 278: What is the relation between transgenic plants used figure 4A-B and 4C-D? Is the result of A-B derived from only one transgenic line? It must be described in the materials, results and figure legend.

We apologize for the lack of clarity. In Figure 4C, D, T3 35S:C4 plants from lines L5 and L7 were used; in Figure 4A, B, and E, due to a limitation in the availability of biological material, we used T2 35S:C4 seedlings from line L5. This has now been indicated in the Methods section as well as in the figure legend.

Reviewer 2 Report

Whereas the importance of  receptor kinases (RLKs) in bacteria and fungi is well documented, their involvement in plant virus interactions is still an open issue. The paper present new results on RLKs in relation with plant viruses. The same group has previously detected an interaction between the C4 protein encoded by TYLCV and the kinase domain of BAM1 RLK, and the results are consistent with the involvement of BAM1 in the C4-mediated suppression of cell-to-cell spread of silencing (PNAS,2018; 115, 1388).

Using the same model virus, the objective was to test if C4 can interact with other RLKs among the large RLK diversity identified in Arabidopsis. Interactions were detected with RLKs of different families. Using Arabidopsis transgenic C4 expressing lines, the biological relevance of these interactions was tested with two C4 interacting RLKs. As with BAM1, some but not all readouts of RLKs pathways were affected. Hence, as in their previous paper on BAM1, the question of the molecular mechanism that determine the interference of C4 in RLKs pathways is still pending.

The paper is very well written.  

Minor comments:

Line 42: should it be ECD instead of EDC?

Line 74 a closing bracket is missing

Line 170-173: the list of eight RLKs with different bracket levels is difficult to read and may be presented in a more readable way. The sentence may be split.

Lines 179-181: Only 7 RLKs listed but 8 were selected (line170). Is there one missing in the list?

Fig. 1: The distance tree should be completed with a scale. The C4-interacting RLKs may be highlighted.

Lines 274-275: please specify reference number for an easy tracking

Line 279: full stop after (B)

Line 288: n=12. What is n? number of leaves?

Line 298: ref 7? please specify for an easy  tracking

Author Response

We would like to thank the reviewers for their critical assessment of our manuscript, and provide a point-by-point response to their comments below.

REVIEWER 2

Whereas the importance of receptor kinases (RLKs) in bacteria and fungi is well documented, their involvement in plant virus interactions is still an open issue. The paper present new results on RLKs in relation with plant viruses. The same group has previously detected an interaction between the C4 protein encoded by TYLCV and the kinase domain of BAM1 RLK, and the results are consistent with the involvement of BAM1 in the C4-mediated suppression of cell-to-cell spread of silencing (PNAS,2018; 115, 1388).

Using the same model virus, the objective was to test if C4 can interact with other RLKs among the large RLK diversity identified in Arabidopsis. Interactions were detected with RLKs of different families. Using Arabidopsis transgenic C4 expressing lines, the biological relevance of these interactions was tested with two C4 interacting RLKs. As with BAM1, some but not all readouts of RLKs pathways were affected. Hence, as in their previous paper on BAM1, the question of the molecular mechanism that determine the interference of C4 in RLKs pathways is still pending.

The paper is very well written.  

Minor comments:

Line 42: should it be ECD instead of EDC?

We thank the reviewer for spotting this typo, which has now been corrected.

Line 74 a closing bracket is missing

We thank the reviewer for spotting this mistake, which has now been corrected.

Line 170-173: the list of eight RLKs with different bracket levels is difficult to read and may be presented in a more readable way. The sentence may be split.

We appreciate this suggestion. We have now re-phrased this list and split the sentence in two to improve readability (lines 172-175).

Lines 179-181: Only 7 RLKs listed but 8 were selected (line170). Is there one missing in the list?

We thank the reviewer for spotting this omission; indeed, CLV1 had to be included. This mistake has now been corrected.

Fig. 1: The distance tree should be completed with a scale. The C4-interacting RLKs may be highlighted.

We thank the reviewer for this comment. With the cladograms in Figure 1, we aimed to represent a hypothetical evolutionary relationship (clades) between different RLKs based on their protein sequence (full or cytoplasmic domain) without necessarily giving information about the evolutionary time history of these relationships. Therefore, in this specific case, the length of the lines does not represent evolutionary time. Following the reviewer’s advice, we have now included a cladogram of the tested RLKs, indicating positive or negative interactions, in Supplementary figure 2.

Lines 274-275: please specify reference number for an easy tracking

We have now added the reference number.

Line 279: full stop after (B)

We thank the reviewer for spotting this mistake, which has now been corrected.

Line 288: n=12. What is n? number of leaves?

“n” here refers to leaf discs (which indeed comes from different leaves). This is now specified in the figure legend.

Line 298: ref 7? please specify for an easy  tracking

We have now added the reference number.